# Enzymatic Spermine Metabolites Induce Apoptosis Associated with Increase of p53, caspase-3 and miR-34a in Both Neuroblastoma Cells, SJNKP and the N-Myc-Amplified Form IMR5

**DOI:** 10.3390/cells10081950

**Published:** 2021-07-31

**Authors:** Yuta Kanamori, Alessia Finotti, Laura Di Magno, Gianluca Canettieri, Tomoaki Tahara, Fabio Timeus, Antonio Greco, Paola Tirassa, Jessica Gasparello, Pasquale Fino, Carlo Maria Di Liegro, Patrizia Proia, Gabriella Schiera, Italia Di Liegro, Roberto Gambari, Enzo Agostinelli

**Affiliations:** 1Department of Biochemical Sciences “A. Rossi Fanelli”, Sapienza University of Rome, Piazzale Aldo Moro 5, 00185 Rome, Italy; guidepost.new-dog@live.jp; 2Department of Life Sciences and Biotechnology, Biochemistry and Molecular Biology Section, University of Ferrara, 44121 Ferrara, Italy; alessia.finotti@unife.it (A.F.); jessica.gasparello@unife.it (J.G.); gam@unife.it (R.G.); 3Department of Molecular Medicine, Sapienza University of Rome, Viale Regina Elena 291, 00161 Rome, Italy; laura.dimagno@uniroma1.it (L.D.M.); gianluca.canettieri@uniroma1.it (G.C.); 4Istituto Pasteur, Fondazione Cenci-Bolognetti, Sapienza University of Rome, Viale Regina Elena 291, 00161 Rome, Italy; 5Department of Sensory Organs, Sapienza University of Rome, Policlinico Umberto I, Viale del Policlinico 155, 00161 Rome, Italy; antonio.greco@uniroma1.it (A.G.); yuteimiyao035391@gmail.com (T.T.); 6Paediatric Onco-haematology, Regina Margherita Children’s Hospital and Paediatric Department, Chivasso Hospital, 10034 Turin, Italy; fabio.timeus@unito.it; 7Institute of Biochemistry and Cell Biology, Research Council of Italy (CNR), 00161 Rome, Italy; paola.tirassa@cnr.it; 8UOC of Dermatology, Policlinico Umberto I Hospital, Sapienza Medical School of Rome, Viale del Policlinico 155, 00161 Rome, Italy; pasquale.fino@gmail.com; 9Department of Biological, Chemical and Pharmaceutical Sciences and Technologies (Dipartimento di Scienze e Tecnologie Biologiche, Chimiche e Farmaceutiche) (STEBICEF), University of Palermo, 90128 Palermo, Italy; carlomaria.diliegro@unipa.it (C.M.D.L.); gabriella.schiera@unipa.it (G.S.); 10Department of Psychology, Educational Science and Human Movement (Dipartimento di Scienze Psicologiche, Pedagogiche, dell’Esercizio fisico e della Formazione), University of Palermo, 90128 Palermo, Italy; patrizia.proia@unipa.it; 11Department of Biomedicine, Neurosciences and Advanced Diagnostics, University of Palermo, 90127 Palermo, Italy; italia.diliegro@unipa.it; 12International Polyamines Foundation ‘ETS-ONLUS’ Via del Forte Tiburtino 98, 00159 Rome, Italy

**Keywords:** polyamine, neuroblastoma, apoptosis, microRNA, mitochondria, reactive oxygen species, oncotherapy

## Abstract

Neuroblastoma (NB) is a common malignant solid tumor in children and accounts for 15% of childhood cancer mortality. Amplification of the N-Myc oncogene is a well-established poor prognostic marker in NB patients and strongly correlates with higher tumor aggression and resistance to treatment. New therapies for patients with N-Myc-amplified NB need to be developed. After treating NB cells with BSAO/SPM, the detection of apoptosis was determined after annexin V-FITC labeling and DNA staining with propidium iodide. The mitochondrial membrane potential activity was checked, labeling cells with the probe JC-1 dye. We analyzed, by real-time RT-PCR, the transcript of genes involved in the apoptotic process, to determine possible down- or upregulation of mRNAs after the treatment on SJNKP and the N-Myc-amplified IMR5 cell lines with BSAO/SPM. The experiments were carried out considering the proapoptotic genes Tp53 and caspase-3. After treatment with BSAO/SPM, both cell lines displayed increased mRNA levels for all these proapoptotic genes. Western blotting analysis with PARP and caspase-3 antibody support that BSAO/SPM treatment induces high levels of apoptosis in cells. The major conclusion is that BSAO/SPM treatment leads to antiproliferative and cytotoxic activity of both NB cell lines, associated with activation of apoptosis.

## 1. Introduction

Neuroblastoma is the most frequent solid tumor of the childhood. It represents 6–10% of all pediatric tumors, and its occurrence after five years of age is a very rare event [1,2]. Neuroblastoma cells derive from the embryonic neural crest, and the tumor can be localized in adrenal medulla or in any area of the sympathetic nervous system [3]. Neuroblastoma is biologically and genetically heterogeneous and also has a heterogeneous clinical behavior, ranging from the spontaneous remission to a very aggressive metastatic disease [4,5,6,7]. The International Neuroblastoma Risk Group (INRG) has developed a classification system for neuroblastoma risk stratification based on clinical criteria, including stage, histology, differentiation, ploidy, alterations at chromosome 11q, and amplification of *MYCN* [8]. This classification allows to divide patient groups into favorable or unfavorable subsets. The amplification of *MYCN* oncogene, encoding for the transcriptional factor N-Myc, strongly correlates with advanced stages and poor outcomes [9,10]. Unfortunately, in spite of very aggressive therapeutic approaches (multimodal chemotherapy, surgery, hematopoietic stem cell transplant), the prognosis of the high-risk neuroblastoma subset is still poor, with a five-year survival below 50% [11,12,13]. Thus, while the current therapy is effective in treating the favorable patients, it is not decisive for the unfavorable ones. Therefore, a new therapeutical approach for unfavorable patients is needed [14]. In this regard, a recent novel strategy for anticancer therapy using polyamines is under investigation [1,15].

Polyamines are organic compounds having more than two amino groups, and with a role in several physiological functions, such as DNA synthesis, cellular proliferation, differentiation, and response to abiotic and biotic stresses [16,17,18,19]. Many findings have demonstrated the important properties of these polycations. Among such properties, polyamines appear to take part in the control of the translation elongation procedure through the Ser/Thr kinases implicated in the phosphorylation of translation elongation factors process, in the control of ion channels gating and in the regulation of oxidative processes [20,21,22,23,24,25]. Moreover, these natural organic compounds are able to control gene expression by modifying the DNA and RNA structure. It is worthwhile mentioning that in a previous publication [26], it was described that spermidine performed considerable cardioprotective and neuroprotective reactions, and ruled out stem cell senescence. In addition, an interesting article reported a novel role of these polycations in the preservation of genome integrity via homology-directed DNA repair [27].

Polyamines are substrates of amine oxidases, a class of enzymes that plays an essential role in the catabolism of polyamines. Bovine serum amine oxidase (BSAO, EC 1.4.3.6) belongs to this class of enzymes found in a wide variety of living organisms. This enzyme was characterized as a homodimeric glycoprotein. Each subunit of 85 kDa contains an organic cofactor, 2,4,5-trihydroxyphenylalanine quinone (TPQ), and an inorganic cofactor, the copper ion. The primary amino group of the aminopropyl moiety, such as the natural polyamines, spermine and spermidine, and benzylamine, is mainly favored in the oxidative deamination reaction by BSAO. This copper enzyme produces enzymatic metabolites, hydrogen peroxide and aldehyde(s), or acrolein, by oxidizing spermine (SPM). This reaction induces apoptosis in melanoma, osteosarcoma, colon adenocarcinoma, and other cell lines.

Apoptotic mechanisms are well-investigated and documented by a plethora of publications about microRNA and apoptosis. MicroRNAs (miRs) are endogenous short noncoding molecules (about 19 to 25 nucleotides in length) deeply involved in post-transcriptional regulation of gene expression [28]. In fact, miRs are able to inhibit mRNA translation or increase mRNA at the level of the multiprotein RNA-induced silencing complex (RISC), mostly depending on the extent of miRNA/mRNA complementarity [29]. It has been demonstrated that a single microRNA is able to interact (mainly at the level of the 3′UTR) with several mRNAs (despite with different affinity). In addition, a single mRNA might be targeted and regulated by several microRNAs (despite with different efficiency). Accordingly, it has been proposed that a relevant portion of the protein-coding genome playing key roles in biological or pathophysiological processes (such as differentiation, cell cycle, and apoptosis) is under microRNA regulation. It was therefore expected (and confirmed by a plethora of scientific reports) that alteration of miRNAs regulation might be associated with several human diseases, including cancer [30,31,32]. Several microRNA (mostly onco-suppressor miRNAs) are downregulated in cancer, including neuroblastoma. For instance, Roth et al. [33] identified miR-193b as a miRNA with tumor suppressive properties. They demonstrated that miR-193b is expressed at low levels in neuroblastoma cell lines and primary tumor samples. The introduction of miR-193b mimetics into nine neuroblastoma cell lines, with distinct genetic characteristics, significantly reduces in vitro cell growth independently of risk factors such as p53 functionality or MYCN amplification. Other examples of miRNAs exhibiting tumor-suppressive properties are miR-584-5p, miR-29a, miR-34a, miR-204, and miR-497 [34,35,36,37,38].

The first aim of this study was to determine whether this new approach can be proposed for other tumoral cell systems, such as neuroblastoma cancer cells, therefore extending and further validated previous works [1,15]. Furthermore, we considered novel approaches focusing on the study of changes of gene expression, including the analysis of the expression of a microRNA associated to the apoptotic pathway. To these aims, two neuroblastoma cell lines, SJNKP and the N-Myc-amplified IMR5 cells, were analyzed. These cell lines are usually used as a model in studies aimed at linking anticancer effects to activation of the apoptotic pathways [39,40,41,42]. In this study, we showed that treatment with BSAO/SPM induced apoptosis in NB cells, via activation of p53, increased miRNA 34a expression, and also promoted mitochondrial membrane depolarization in the N-Myc-amplified neuroblastoma cell line. These findings suggest that the proposal of a novel therapeutic approach using the combinatorial treatment with BSAO/SPM may be taken into account. This approach might represent a potential therapy to defeat NB cancer in unfavorable patients.

## 2. Materials and Methods

### 2.1. Reagents

Spermine tetrahydrochloride, fetal bovine serum (FBS), ribonuclease A (RNase A), thiazolyl blue tetrazolium bromide (MTT) and Tri-Reagent, propidium iodide (PI), nuclease free water, ethanol, 5,5’,6,6’-tetrachloro-1,1’,3,3’-tetraethyl-imidacarbocyanine iodide (JC-1), RPMI-1640 medium with l-glutamine and sodium bicarbonate, and methyl α-d-mannopyranoside were purchased from Sigma-Aldrich (St. Louis, MO, USA). Annexin V-FITC apoptosis detection kit was obtained from Enzo Life Sciences (Farmingdale, NY, USA). Primary antibodies against caspase-3, PARP, and tubulin were purchased from Cell Signaling Technology (Danvers, MA, USA). TaqMan Reverse Transcription PCR Kit, TaqMan MicroRNA Reverse Transcription Kit, TaqMan Universal Master Mix, no UNG 2X, and random hexamers were purchased from Thermo Fisher Scientific (Waltham, MA, USA). TaqMan probes and PrimeTime Gene Expression Master Mix 2X were obtained from Integrated DNA Technologies (Coralville, IA, USA). Carboxymethyl cellulose cation-exchange resin was purchased from Whatman (Maidstone, UK). Horseradish peroxidase conjugated secondary antibody was obtained from Bethyl Laboratories.INC (Montgomery, Alabama, USA). Western Bright ECL was obtained from Advansta (San Jose, CA, USA). Concanavalin A-Sepharose, SP-sepharose, and Q-Sepharose were purchased from GE Healthcare (Little Chalfont, UK). All cell culture flasks and dishes were obtained from Corning (Corning, NY, USA).

### 2.2. Purification and Determination of Catalytic Properties of BSAO

Bovine blood was withdrawn at a slaughterhouse and mixed with 3.8% sodium citrate solution (an anticoagulant), and then the enzyme bovine serum amine oxidase (BSAO) was purified by a combination of ionic exchange and affinity chromatographic processes, as previously described [43,44]. All purification steps were carried out in a cold room, at 4 °C. The BSAO purification factor was approximately 1600-fold and a single band was obtained on 6% SDS-PAGE gel.

The BSAO catalytic properties were spectrophotometrically obtained by determining the amount of H_2_O_2_ generated during the oxidation of the spermine or spermidine. In the presence of H_2_O_2_, 3,5-dichloro-2-hydroxybenzene-sulfonic acid (DCHBS) is oxidized by horseradish peroxidase (HRP) to a semiquinone radical form, which forms a pink quinoneimine derivative reacting with 4-aminoantipyrine (AAP). When 1 mM AAP and 2 mM DCHBS were present, the determinations were carried out in 0.1 M PBS buffer (pH 7.4) at 37 °C, using spermine as a substrate to reveal BSAO activity [45]. The pink adduct (ε_515_ = 26,000 M^−1^ cm^−1^), formed by the oxidation of the polyamine, was detected spectrophotometrically. The kinetic constants (affinity constant (*K*m), catalytic constant (*k*cat), and specificity constant (*k*cat/*K*m)) values were estimated using Lineweaver–Burk plots. The protein concentration for the enzyme was determined by the absorbance at 280 nm, applying an absorption extinction coefficient of 1.74 L g^−1^ cm^−1^. A total of 6.5×10^−3^ U/mL of BSAO was added for the experiments carried on cell cultures. BSAO activity was also evaluated as the amount of benzaldehyde generated per min (μmol/min) from benzylamine at 250 nm (ε = 12,500 M^−1^ cm^−1^) in 0.1 M sodium phosphate buffer (pH 7.2) at 25 °C.

### 2.3. Neuroblastoma Cell Culture

As mentioned above, two types of neuroblastoma cell lines (NB) have been used in this study: the first one, named SJNKP, does not harbor any myc amplification, while the second one, named IMR5, harbors N-myc amplification. These cell lines were donated as a gift by Dr. Nicoletta Crescenzio (Department of Paediatrics, University of Turin, Regina Margherita Children’s Hospital, Piazza Polonia 94, I-10126 Turin, Italy) and by Dr. Fabio Timeus. Both SJNKP and IMR5 cells were grown in monolayer cultures in RPMI-1640 medium supplemented by adding 10% fetal bovine serum (FBS), 2 mM L-glutamine, antibiotics penicillin (100 IU/mL), and streptomycin (100 μg/mL) in a humidified atmosphere of 5% CO_2_ in a water-jacketed incubator at 37 °C. For each passage, both cell lines were harvested with 10 mM EDTA in PBS, and then by the further addition of 0.25% trypsin solution dissolved in PBS. The trypsin activity was quenched by the addition of RPMI-1640 medium containing FBS [42].

### 2.4. Isolation of Neurons from Rat Brain Cortex

Wistar rats (Harlan, Udine, Italy) were housed in the animal facilities of the STEBICEF Department, University of Palermo, Palermo, Italy. All used procedures were in agreement with the European Community Council Directive and approved by a licensed veterinary surgeon, and by the Animal Welfare Committee of the University of Palermo; the experiments were also authorized by the Ministry of Health (Rome, Italy; authorization number 69636.N.GCQ). The number of animals used was minimized as much as possible. Neurons were purified from rat brain cortex at the 16th day of gestation and cultured at the bottom of laminin-coated (2.5 μg/cm^2^) wells, in the serum-free Maat Medium [46,47]. In detail, cerebral hemispheres from 16-day-old rat fetuses were removed by aseptic surgical procedure, placed in Dulbecco–Vogt modification of Eagle’s medium (DME) +20% NCS (newborn calf serum), and cleaned of their meningeal coverings under dissecting microscope. The tissue was fragmented and dissociated as previously described by Caradonna et al. [48]. After centrifugation at 300 *g* for 5 min, cells were resuspended in DME, washed twice, and finally resuspended in Maat Medium and then plated at 10^6^ cells/cm^2^.

### 2.5. Treatments, Clonogenic and MTT Cytotoxicity Assays

#### 2.5.1. Clonogenic Assay

The cytotoxicity caused by polyamine metabolites, H_2_O_2_ and aldehyde(s), on both SJNKP and IMR5 cell lines was determined by using a plating clonogenic assay. Cell viability assays were performed on subconfluent cells that had been incubated in fresh medium, supplemented with FBS, for 24 h at 37 °C, before each treatment. Cells were detached by addition of 10 mM EDTA dissolved in phosphate-buffered saline (PBS) and then of 0.25% trypsin in PBS. The harvested cells were washed with PBS containing 1% bovine serum albumin (BSA) (PBS-1% BSA), and pelleted by centrifugation at 1500 rpm for 2 min. The cells were resuspended in PBS-1% BSA. Aliquots of freshly harvested NB cells (10^5^/mL) were incubated at 37 °C for different time periods, up to 60 min, with several concentrations of spermine in presence of BSAO (17.20 μg/mL corresponding to 1.01 × 10^−4^ μmoles/mL or 6.5 × 10^−3^ IU/mL), used alone or in combination. SPM solutions were freshly prepared before each experiment, dissolved in water, and, if present, were added last, to initiate the enzymatic reaction. After incubation, cells were washed with PBS-1% BSA twice, centrifuged, and resuspended in 1 mL PBS-1% BSA. The cytotoxic effect was evaluated by a plating efficiency assay, which detects the capacity of the cells to generate macroscopic colonies (≥50 cells). Neuroblastoma cells were subsequently seeded in Petri dishes with RPMI-1640 medium, supplemented with FBS, and incubated at 37 °C and 5% CO_2_ for about two weeks until colonies were reproduced. The colony-forming cells were then washed, stained with methylene blue, and counted according to Calcabrini et al. [49].

#### 2.5.2. MTT Assay

The cytotoxicity caused by H_2_O_2_ and aldehyde(s) on both NB cell lines was also detected by using a MTT assay. Cells were removed from the substrate and, after resuspension in PBS-1% BSA buffer, harvested cells (10^5^/mL) were incubated at 37 °C, for up to 60 min, in the presence of different amount of spermine and BSAO (6.5 × 10^−3^ IU/mL), used alone or in combination, as described in the clonogenic assay section. After incubation, cells were washed with PBS-1% BSA twice, centrifuged, and resuspended in 1 mL PBS-1% BSA. Both SJNKP and IMR5 cells were subsequently seeded in a 96-multiwell plate and processed according to Kanamori et al. [50].

### 2.6. Measurements “In Situ” of Mitochondrial Membrane Potential (Δψ_m_)

The changes in mitochondrial membrane potential (ΔΨm) in neuroblastoma cells, SJNKP and IMR5, were determined using the lipophilic cationic probe, JC-1 dye, according to Calcabrini et al. [49]. SJNKP and IMR5 cell lines were collected using EDTA/trypsin, washed with PBS-1% BSA, centrifuged at 600× *g* for 2 min at 25 °C, and then resuspended in PBS-1% BSA, as reported in the assays above. The cells were treated with different concentrations of substrate SPM in the presence of BSAO (6.5 × 10^−3^ IU/mL) during 60 min at 37 °C. Then, NB cells were marked with 2.5 μg/mL of JC-1 during the final 15 min of treatment at 37 °C. The cells removed from the substrate were washed with cold PBS-1% BSA and then resuspended in cold PBS. The samples were examined using a BD Accuri C6 flow cytometer, as previously reported [49].

### 2.7. Determination of Apoptotic Cell Death by Annexin V-FITC Staining

The annexin V-FITC/PI assay is a Ca^2+^-dependent phospholipid-binding protein which shows a high affinity for phosphatidylserine (PS) residues located on the outer layer of the plasma membrane cells and represents an early signal of the apoptotic process. To determine phosphatidylserine (PS) residues, the assay based on the use of the Annexin V-FITC Apoptosis Detection Kit was performed as previously published by Van Engeland et al. [51]. Neuroblastoma cells, SJNKP, and IMR5 cells (5 × 10^5^/mL) were handled with different concentrations of spermine and BSAO (6.5 × 10^−3^ IU/mL) for 1 h at 37 °C, as reported above in the paragraph about cell viability assay. Cell suspensions were seeded in a 6-well plate in presence of RPMI-1640 medium with glutamine and supplemented with FBS. After incubation at 37 °C for 48 h, NB cells were removed from the substrate, washed with PBS, centrifuged for a few min at 25 °C, resuspended in 100 μL of binding buffer containing 10 mM HEPES/NaOH (pH 7.4, 140 mM sodium chloride, 2.5 mM CaCl_2_), then the cells were stained with 1 μg/mL of annexin V-FITC and with 1 μg/mL of propidium iodide (PI) and incubated for 10 min at 25 °C in the dark. Annexin V-FITC and PI fluorescence were performed on the FL-1 channel (533/30 nm) and the FL-3 channel (>670 nm), respectively, with an excitation at 488 nm. At least 10,000 events/sample were acquired, using a BD Accuri C6 flow cytometer (BD Biosciences, San Jose, CA, USA).

### 2.8. Cell Cycle Analysis

Cell cycle analysis was carried out by labeling neuroblastoma cells with PI. The assays were performed as previously reported by Nicoletti et al. [52]. Spermine and BSAO-treated and untreated neuroblastoma, SJNKP, and IMR5 cells (5 × 10^5^/mL) were treated with BSAO and SPM as described in the annexin V-FITC/PI paragraph. After the reaction, the pellet was fixed with cold ethanol 70% for 60 min in cold room at 4 °C. After washing with PBS, the cells were centrifugated and resuspended in PBS containing 100 μg/mL RNase A and 40 μg/mL PI. Following incubation at 37 °C for 60 min, NB cells were then analyzed by flow cytometry using the FL-3 channel (>670 nm) with the acquisition of 10,000 events/sample.

### 2.9. RNA Isolation and Analysis

Frozen cellular pellets were defrosted and lysed with 1 mL of cold TriReagent, at a ratio of 1 mL of TriReagent for 1 × 10^6^ cells. The extraction was performed according to manufacturer’s instructions, briefly: samples were incubated for 5 min at room temperature (RT), then 200 µL of nuclease-free chloroform was added. The mixture was shaken vigorously and incubated at RT for 3 min. Samples were centrifuged at 12,000 rpm for 10 min in order to obtain phase separation, and the aqueous phase was taken and transferred to a new collection tube. A total of 500 µL of nuclease-free isopropanol was added, and the mixture was incubated at room temperature for 10 min and then centrifuged at 12,000 rpm for 15 min to precipitate RNA. Supernatant was removed and RNA was washed with 1 mL of cold 75% ethanol, air dried, and resuspended in nuclease-free water. Obtained RNA was quantified using SmartSpec plus Spectrophotometer (Bio-Rad, Hercules, CA, USA).

### 2.10. mRNA Reverse Transcription and Quantitative Real Time PCR (RT-qPCR)

A total of 500 ng of total RNA was reverse transcribed using Taq-Man Reverse Transcription PCR Kit and random hexamers. A mixture containing total RNA and nuclease-free water was heated at 60 °C for 5 min to disrupt RNA secondary structure. After the incubation, a mixture composed of RNase inhibitor enzyme at a final concentration of 0.4 U/µL and random hexamers, at a final concentration of 2.5 µM, was added, and the samples were incubated for 10 min at 20 °C to allow primer annealing. At the end of the primer annealing step, other reagents necessary for reverse transcription reaction were added, including dNTPs at a final concentration of 2 mM, MgCl_2_ (magnesium chloride) 5.5 mM, RT Buffer 1× and MultiScribe Reverse Transcriptase 1.25 U/µL. The mixture was incubated at 42 °C for 30 min to allow mRNA reverse transcription, followed by a step at 98 °C for 5 min to inactivate reverse transcriptase enzyme. RT-qPCR was performed using TaqMan probes. Table 1 shows gene names and probe sequences. Briefly, a PCR reaction mix composed of 1 µL of cDNA, 1.25 µL of 20× TaqMan assay for the transcript of the interest, 12.5 µL of IDT Master Mix 2×, and nuclease-free water at a final volume of 25 µL was prepared. Each sample was prepared and loaded in duplicate. The amplification program consists of an initial incubation at 95 °C for 3 min to activate DNA polymerase enzyme, and 40 cycles in which the sample is heated at 95 °C for 15 s to denature double-strand structure, followed by a step at 60 °C for 1 min in which primer annealing and extension phase occurs. At the end of each cycle, fluorescence is measured. The amplification was performed using a CFX96 Touch Real-Time PCR Detection System. Two pro-apoptotic transcripts were analyzed: Tp53 and caspase-3 which are, respectively, at the beginning and at terminal phases of the mitochondrial apoptosis pathway [53]. Relative expression was calculated using the comparative cycle threshold method (2^−ΔΔCT^),and the endogenous control human RPL13A was used as normalizer gene. Duplicate negative controls (no template cDNA) were also run with every experimental plate to assess specificity and to rule out contamination. All RT-qPCR reactions were performed in triplicate for both target and normalizer genes.

### 2.11. miRNA Reverse Transcription and Quantitative Real Time PCR (RT-qPCR)

A total of 300 ng of total RNA was reverse transcribed in a final volume of 30 µL using the TaqMan MicroRNA Reverse Transcription Kit and specific miRNA primers (assay ID are reported in Table 2), according to the manufacturer’s protocol. A total of 3 µL of obtained cDNA was amplified for 50 PCR cycles using the TaqMan Universal Master Mix, no UNG 2X, and specific TaqMan 20x miRNA assay for miR-34a-5p or for the housekeeping, miR-let-7c and U6 snRNA. The following amplification program was employed: 95 °C for 10 min to activate Taq polymerase enzyme, 95 °C for 15 s to denature cDNA double-strand structure, followed by a step at 60 °C for 1 min to allow primers and probe annealing. Amplification was performed using the CFX96 Touch Real-Time PCR Detection System (Bio-Rad), and relative expression was calculated using the comparative cycle threshold method (2^−ΔΔCT^).

### 2.12. Western Blotting Analysis

Cleavage of caspase-3 and of poly-ADP-ribose polymerase (PARP, a major substrate of caspase-3) is considered a valuable marker of apoptosis. Caspase-3 is a member of the caspase family that comprises 13 proteases that play a central role in the activation of the apoptotic program. The activation of caspase-3 mediates apoptotic cell death through the downstream cleavage of several key cytoplasmic or nuclear substrates and is primarily responsible for the cleavage of PARP that signals DNA damage leading to cell death. Neuroblastoma cells, SJNKP and IMR5, were collected and lysed in RIPA buffer (50 mM Tris-HCl pH 7.6; 0.5% sodium deoxycholate; 140 mM NaCl; 1% NP40; 5 mM EDTA pH 8.0; 100 mM NaF, 2 mM sodium pyrophosphate) containing protease inhibitors (40 mg/mL pepstatin, leupeptin, and aprotinin, 0.5 mM PMSF). Cellular extracts were then centrifuged at 13,000 rpm at 4 °C for 30 min, the supernatant was collected, and protein content was quantified by Bradford assay (Bradford Reagent, Bio-Rad, Hercules, CA, USA). Then, equal amounts of samples were diluted in sample buffer (1M Tris-HCl pH 6.8, 10% glycerol, 6% SDS, 10% β-mercaptoethanol, 0.2% bromophenol blue in EtOH), boiled for 5 min, and separated by SDS-PAGE. Proteins were then transferred onto a nitrocellulose membrane. Membranes were blocked in 5% skim milk in TBS-T (Tris-Base 20 mM, NaCl 0.1 M pH 7.5, 0.05% Tween-20) for 30 min, and incubated with primary antibodies against cleaved caspase-3 (#9661S, Cell Signaling Technology), PARP (#9542S, Cell Signaling Technology), and α-tubulin (#sc-8035, Santa Cruz Biotechnology), with gentle shaking at 4 °C overnight. The membranes were then incubated with horseradish peroxidase-conjugated secondary antibodies at room temperature for 30 min. After incubation, filters were washed with TBS-T and exposed to Western Bright ECL solution (Advansta, #K-12045-D50) for 5 min, and chemiluminescence was detected through an Azure c600 ChemiDoc Imaging System (Azure Biosystems, Dublin, CA, USA). For densitometry analysis, signal intensity was quantified by Image J software (Rasband, W.S., ImageJ, U.S. National Institutes of Health, Bethesda, MD, USA) [54].

### 2.13. Vitality Assay

Primary neurons and both neuroblastoma cell lines were treated with BSAO/SPM for 1 h at 37 °C, washed twice, and then incubated at 37 °C for 48 h on round glasses for microscopy. With the aim to recognize an apoptotic and/or a necrotic process on cells, the physiological status of cells was detected by acridine orange/ethidium bromide (AO /EtBr) staining, and by fluorescence microscopy using an Olympus apparatus BX-50. Briefly, cells were washed with PBS and stained with AO/EtBr solution, using for each compound a concentration of 100 μg/mL in PBS. Fluorescence images were taken in random fields, and calculating of viable cells was carried out after dividing each image into quarters.

### 2.14. Statistical Analysis

The data were expressed as mean ± SD. The statistical significance of differences was determined using one-way analysis of variance (ANOVA) followed by Dunnett’s multiple comparison test. Differences at *p* < 0.05 were considered to be significant. All statistical analyses were performed using the EZR [55].

## 3. Results

### 3.1. Cytotoxic Effect Induced by BSAO in the Presence of SPM on Neuroblastoma Cell Lines

In order to verify the possible cytotoxic effect induced by the oxidation products of spermine, H_2_O_2_ and aldehydes, resulting from the enzymatic reaction catalyzed by BSAO, both neuroblastoma cell lines, IMR5 and SJNKP, were treated with increasing concentrations of spermine in presence of BSAO**.** Cell viability was evaluated by a clonogenic assay (Figure 1A,B). Figure 1 shows the results of this assay, confirming that the treatment with BSAO alone was unable to cause cytotoxic effects. The lack of cytotoxic effects of SPM alone is firmly established in these cell lines and previously reported by Agostinelli et al. [56]. Instead, when SPM was added in the presence of BSAO, a concentration-dependent cytotoxic effect was observed, which was already detectable in the presence of 6 µM SPM concentration for both cell lines.

To support this result, an MTT assay was also performed to evaluate the cytotoxic effects induced by BSAO/spermine enzymatic system (Figure 1C,D). As shown in Figure 1C,D, the cytotoxic effect was comparable with that observed by the clonogenic assay, as represented in Figure 1A,B. Moreover, MTT assay confirmed that BSAO alone was not cytotoxic, while BSAO in presence of SPM 6 µM caused cytotoxicity in both cell lines, SJNKP cells being more sensitive to the treatment than IMR5 cells. In fact, while 6 µM SPM reduced, by about 50%, the viability of SJNKP cells, the same concentration resulted in a less pronounced reduction of IMR5 cells viability, where a solution of 12 µM SPM concentration resulted in a stronger effect.

### 3.2. Proapoptotic Effects on Neuroblastoma Cell Lines: Annexin V-FITC/PI Assay and Cell Cycle Analysis by Flow Cytometry

In order to verify whether the cytotoxicity induced by BSAO and exogenous SPM in SJNKP and IMR5 neuroblastoma cell lines was due to activation of the apoptotic pathway, annexin V-FITC/PI double staining assay and cell cycle analysis were performed. The results of annexin V-FITC/PI assay are shown in Figure 2A,C for SJNKP and IMR5 cells, respectively. They demonstrate the lack of proapoptotic activity after the treatment of both neuroblastoma cells, SJNKP and IMR5, with BSAO alone. On the contrary, when added in the same incubation mixture, BSAO and 6 µM SPM were able to induce apoptosis in both cell lines. The effects of 6 µM SPM on BSAO treated cells (Figure 2A,B) exhibit the same trend found in the cytotoxic assays described in Figure 1.**** SJNKP cells resulted as more susceptible to the apoptosis induction than IMR5 ones. A high apoptotic effect (an induction higher than 60% of early apoptotic cells) was determined for both cell lines using concentrations of SPM 9 and 12 µM for SJNKP and IMR5 cells, respectively. Moreover, concentrations of SPM 18 µM showed, in IMR5, the highest cell death, corresponding to 82.5% of early apoptotic cells. The percentage increase of cells in early apoptosis, in function of spermine concentration, is mainly due to the high concentration of H_2_O_2_ produced by the enzymatic reaction, during the first 10 min of the reaction (approximately 80%). It is able to cross the cell membranes, as already determined in LoVo colon adenocarcinoma cells [49]. In Figure 2B,D are shown quantitative analyses of percentage of normal and apoptotic cells in SJNKP and IMR5 cell lines, respectively. To support the involvement of the apoptotic process caused by BSAO/spermine, flow cytometric analysis using PI staining was carried out to analyze the cell cycle status. The apoptotic cells that undergo DNA fragmentation exhibit sub-G1 DNA contents. The cytotoxic effects induced by BSAO/SPM enzymatic system on cell cycle, in both NB cell lines, are shown in Figure 3A,C for SJNKP and IMR5 cells, respectively. They demonstrate a strong effect of BSAO/SPM treatment on the accumulation of the sub-G1 cell fraction. This assay also showed that 6 µM SPM was able to induce sub-G1 cell accumulation in both cell lines. SJNKP cells appeared slightly more susceptible to this increase of induction due to the treatment. The main point of this analysis is that the proportion of sub-G1 cells was found higher than 60% in SJNKP and more than 75% in IMR5 cells when treated with BSAO and SPM 9 µM and 18 µM, as shown in Figure 3B,D, respectively. In Figure 3B,D are reported quantitative analysis of sub-G1 phase of both SJNKP and IMR5 cell lines, respectively. These results confirm the cytotoxicity induced by spermine enzymatic oxidation products, H_2_O_2_ and aldehydes, and suggest the activation of the apoptotic pathway.

### 3.3. Effect of BSAO and SPM on Mitochondrial Membrane Potential in Neuroblastoma Cells

To study the mechanisms through which spermine metabolites, H_2_O_2_ and aldehydes, formed by BSAO during the enzymatic reaction, caused cell death, flow cytometric analysis was performed on the control and treated neuroblastoma cell lines, loaded with the mitochondrial probe JC-1, to follow the loss of mitochondrial membrane potential (ΔΨm). JC-1 dye is used as an indicator of ΔΨm in a variety of tumor cell types. In healthy cells, its cationic carbocyanine dye spontaneously accumulates in the mitochondria and forms J-aggregates able to emit red fluorescence after excitation. Instead, in apoptotic cells, after the treatment, which shows a low value of ΔΨm due to mitochondrial membrane depolarization (MMD), JC-1 remains in the cytoplasm as monomers, and reversibly changes color emitting green fluorescence. Therefore, the loss of ΔΨm is indicated by a decrease in the ratio of red/green fluorescence intensity. Thus, to clarify whether the apoptotic process induced by BSAO/SPM, in both neuroblastoma cell lines, is related to mitochondria dysfunctions, JC-1 dye was used to detect any MMD.

As shown in Figure 4, in absence or in presence of BSAO alone, significant differences in MMD were not observed in both neuroblastoma cell lines. Instead, in the SJNKP and IMR5 cells, the exposure to spermine 9 µM or 18 µM, respectively, in the presence of BSAO, caused an evident MMD in a dose-dependent manner. The spermine metabolites, H_2_O_2_ and aldehydes, generated by the enzymatic reaction, induced a significant decrease in the ratio of red/green fluorescence intensity when compared to control neuroblastoma cell samples. Therefore, these findings suggest that BSAO/SPM treatment was able to induce an evident ΔΨm dissipation, at least when the highest doses of SPM were employed, since the enzymatic system increased the green fluorescence intensity, accompanied by a decrease in the intensity of red fluorescence in both neuroblastoma cell lines.

### 3.4. Analysis of Mitochondrial Apoptosis-Related mRNAs

To further confirm previously presented data, we analyzed the transcripts of some genes involved in the apoptotic process by real-time RT-qPCR. Experiments were carried out considering the proapoptotic genes p53**,** involved in the first stages of the apoptotic pathway, and caspase-3, involved in the terminal phase of apoptotic process. After the treatment with BSAO and SPM 6 µM and 9 µM, the SJNKP cells displayed a dose-dependent increase of the levels of the mRNAs encoded by the two proapoptotic genes. In agreement, also in the IMR5 cell line, the treatment with BSAO and SPM was able to induce an increase of the expression of these proapoptotic genes, but in this case, higher concentrations of SPM (12 µM and 18 µM) were required, while no significant variations in transcripts content were detected when lower concentrations of SPM (6 µM) were employed. Interestingly, both analyzed transcripts (p53 and caspase-3) were increased dose-dependently, as shown in Figure 5.

### 3.5. Analysis of Apoptosis-Related miR-34a

Previous studies showed that miR-34a-5p is a key tumor suppressor microRNA, whose downregulation has been strongly linked to growth of neuroblastomas [57]. MiR-34a inhibits silent information regulator 1 (SIRT1) expression through a miR-34a-binding site within the 3’ UTR of SIRT1. MiR-34a inhibition of SIRT1 leads to an increase in acetylated p53. Moreover, miR-34a is a direct target gene of p53. Therefore, the miR-34a/SIRT1/p53 signaling pathway forms a positive feedback loop wherein p53 induces miR-34a and miR-34a activates p53 by inhibiting SIRT1. Since we found that p53 mRNA levels are upregulated by SPM/BSAO treatment, and since miR-34a-5p is a target of p53, we next wondered whether this microRNA is also modulated by SPM/BSAO treatment. To this end, we tested the effect of BSAO/SPM on miR-34a by performing RT-qPCR.

The effects of SPM/BSAO treatment on miR-34a are shown in Figure 6. No major changes in miR-34a content were found when SJNKP and IMR5 were treated with only BSAO. On the contrary, a significant increase (*p* < 0.05 or *p* < 0.01) of miR34a content was observed when both cell lines were treated with BSAO and SPM at the 6, 12, and 18 µM concentrations. According to data obtained by analyzing p53 transcript, the effects on intracellular miR-34a content are dose-dependent (Figure 5).

### 3.6. Effect of BSAO and SPM on Proapoptotic Protein Expression

In order to further investigate the mechanism of apoptosis induced by BSAO/SPM, Western blotting analysis was performed to detect cleaved caspase-3 and cleaved PARP protein expression levels (Appendix A).

Following treatment with BSAO and 36 µM SPM, intensity of cleaved caspase-3 and cleaved PARP signal was significantly increased in SJNKP cell line (Figure 7A,B). Moreover, the same effect was confirmed in IMR5 cell line treated with BSAO and 36 µM SPM (Figure 7C,D). Quantification (relative density) of the intensity of staining of full-length PARP and cleaved PARP and caspase-3 protein from SJNKP and IMR5 cell lines are shown in Figure 7B,D, respectively. These results suggest that BSAO/SPM treatment induces not only mRNA synthesis, but also protein expression of proapoptotic genes.

### 3.7. Comparison of the Effects of BSAO/SPM on Both Neuroblastoma and Neurons

In order to check if there is any difference between neurons and neuroblastoma cells treated with BSAO/SPM, dual AO/EB staining was used. AO is a vital dye able to stain both live and dead cells, while EB staining only allows the detection of cells that have lost membrane integrity. In this study, dual AO/EB staining was used to examine cell death in two neuroblastoma cell lines and primary neurons treated with BSAO/SPM.

Figure 8 shows that BSAO and 18 µM SPM induced cell death in both neuroblastoma cell lines, while it had no effect on neurons. In addition, there was no significant difference between neurons and neuroblastoma cells treated with 0, 6, and 54 µM SPM in the presence of BSAO. Evaluation of cell viability by ImageJ showed that there are significant differences between neurons and neuroblastoma cells treated with 18 µM of SPM and BSAO. As a whole, these results suggest that primary neurons are more resistant to the cytotoxic effects induced by hydrogen peroxide and acrolein than neuroblastoma cancer cells, and highlight the specificity and potential safety of the treatment against normal, noncancerous tissues.

## 4. Discussion

Polyamines are involved in several physiological processes, such as DNA synthesis, signal transduction, cell proliferation, and differentiation, and dysregulation of polyamine metabolism promotes the progression of cancer. In fact, polyamine concentrations are increased in multiple solid tumors due to both overexpression of ornithine decarboxylase [58,59] and the polyamine transport system [60]. Therefore, it is expected that inhibitors of polyamine metabolism could be meaningful anticancer drugs for chemotherapy.

Anticancer drugs should exhibit selective tumor-targeting effects, avoiding major side effects on normal cells. Moreover, the biochemical and molecular basis of the drugs might indicate possible mechanisms of action allowing identification of novel therapeutic targets, which are expected to facilitate personalized approaches on one hand, and multiple treatments on the other. In this respect, the use of drug combination at low drug concentration, or therapeutic systems such as chemotherapy and hyperthermia, might importantly reduce side effects. In this regard, our research group has previously demonstrated by clonogenic assay that multidrug-resistant (MDR) colon adenocarcinoma cells (LoVo DX) are more sensitive than the corresponding wild type cells (LoVo WT) to H_2_O_2_ and aldehyde(s). Both cytotoxic metabolites were responsible for the loss of cell viability. Transmission electron microscopy observations showed that BSAO and SPM induced evident mitochondria alterations, more pronounced in LoVo DX than in LoVo WT cells. The mitochondrial activity was checked by flow cytometry studies, labeling cells with the probe JC1 dye. After treatment with BSAO and SPM, the cells showed a marked increase in MMD, higher in LoVo DX than in LoVo WT cells [15,49,61].

In the present study, we investigated the possible use of BSAO as an anticancer agent in the presence of SPM. Spermine is deaminated by BSAO, producing hydrogen peroxide and the corresponding aldehyde(s). According to previous studies on BSAO, these reactive compounds, formed in the presence of polyamine, could induce cytotoxicity on several cancer cell lines in vitro, or when generated by endogenous polyamines, and may further cause depletion of polyamines in the tumor microenvironment in vivo, resulting in growth inhibition. However, the mechanism of action induced by the enzymatic system BSAO/SPM is still under investigation.

This experimental study was first carried out in vitro to examine cytotoxicity and cell growth inhibition induced by polyamine metabolites, generated by BSAO, on both NB cell lines, N-myc-nonamplified SJNKP, and N-myc-amplified IMR5.

As the effectiveness of therapeutic drugs is conditional on their long-term action on tumor cells, the result of the treatments on in vitro tumorigenic ability was studied using a clonogenic assay, which is recognized to be the most rational method for the estimation of cancer cell sensitivity to antitumor agents, and an MTT assay, which is frequently used because it reflects the number of viable cells under defined conditions. In previous studies, as reported above, it was showed that exogenous spermine metabolites, H_2_O_2_ and aldehyde(s), can produce a cytotoxic effect, higher in both phenotypes of multidrug-resistant LoVo DX and M14 ADR2 human cancer cells than in their wild-type counterparts [1,49]. Indeed, consistent with previous observations, exogenous spermine and BSAO treatment significantly reduced cell viability in an SPM dose-dependent manner in both NB cell lines (Figure 1). Interestingly, these results showed that BSAO/SPM treatment exhibits a cytotoxic effect on NB cell lines in spite of N-myc overexpression.

Annexin V and cell cycle assays showed that treatment with spermine and BSAO significantly increased the percentage of apoptotic cells in a spermine dose-dependent manner (Figure 2 and Figure 3), indicating that the treatment with the enzymatic systems is able to induce cell death through an apoptotic pathway. The involvement of apoptosis was also detectable in both LoVo and M14 cells wild-type and MDR [1,15,49].

It is known that there are two major pathways of apoptosis: the extrinsic pathway and intrinsic ones. The extrinsic pathway is triggered by binding of Fas ligand or TNF with their cognate receptors, recruiting procaspase-8 and activating downstream effectors caspase-3 and/or caspase-7 [62]. Instead, the intrinsic pathway is identified by the mitochondrial outer membrane permeabilization and the freeing of caspase activators, with cytochrome C and including other proapoptotic compounds into the cytoplasm, supporting caspase-9 activation. Activated caspase-9, in turn, conducts cleavage of the executioner caspase-3 and caspase-7, which subsequently activates the final events of programmed cell death [63]. Therefore, mitochondria have an essential part in intrinsic apoptosis pathway, and a decrease in ΔΨm is an early irreversible step of the process [15,64]. The results obtained in this research suggest that spermine metabolite-caused apoptosis is anticipated by ΔΨm collapse in both SJNKP and IMR5 NB cell lines (Figure 4). Hydrogen peroxide generated by the oxidation of spermine in the extracellular environment is able to cross both cellular membranes, and the inner membrane of mitochondria, and directly interacts with endogenous molecules and structures, inducing an intense oxidative stress under appropriate conditions [15,49,65,66]. The stress provoked by the enzymatic oxidation products of spermine is implicated in triggering of the mitochondrial permeability transition process that permits opening of the transition pore, conducting loss of ΔΨm and mitochondrial dysfunctions, as well as mitochondrial swelling and rupture of the outer membrane. In addition, it was observed in a previous study that a concentration of 12 μM of commercial H_2_O_2_, or produced by the enzymatic reaction, BSAO/SPM, caused an increment in mRNA levels for the BAX proapoptotic gene in adenocarcinomas multidrug-resistant LoVo DX cells, while the pro-survival Bcl-2 gene did not expose any difference after the treatment with hydrogen peroxide [67].

As we have demonstrated in this experimental work, BSAO/SPM treatment induced mitochondrial-related apoptosis. However, there are still several points that need further investigation. In an intrinsic pathway, several factors, such as p53, caspase-3, and PARP, have an important role. Moreover, several papers report that miR-34a is involved in the apoptotic process and that it is, in particular, related to p53 [68]. This working hypothesis is shown in Figure 9. In this scheme, miR-34a might play a crucial role, leading to downmodulation of SIRT-1 and BCL-2 [69], two inhibitors of p53 gene transcription. Activation of p53 leads, in turn, to at least two effects: (a) a further potentiation of miR-34a expression and (b) activation of apoptosis and caspase-3. This working hypothesis is sustained by the results shown in Figure 5, Figure 6 and Figure 7 and allows to propose miR-34a mimicking as a strategy to potentiate antitumor effects of anticancer drugs, including BSAO/SPM treatment. In this respect, combined treatments using modulators of miRNA biological activity and antitumor agents have been proposed as an important strategy to maximize antitumor effects (including activation of apoptosis) [30,70]. We would like to underline that the BSAO/SPM effects occurring at different concentrations of reagents are largely expected in the different cellular assays employed (compare, for instance, the experiments shown in Figure 7 with those described in Figure 2 and Figure 3); still, it is worth noting a similar trend was observed.

When studying the potential usefulness of a given treatment, it is of the most importance to compare the effects of the compounds under study on both cancer and normal cell. In this study, we performed experiments to verify the selectivity of the BSAO/SPM treatment for tumor cells excluding toxic phenomena on normal cells, representative of healthy tissues. Anticancer possibilities of the enzyme BSAO have been previously demonstrated by performing experiments in vivo using a mouse melanoma model [71,72]. It is worth remembering that, in the development of new approaches in cancer therapy, their side effects against normal cells should be taken into consideration. An interesting study has shown that normal human epidermal melanocytes exhibit a higher resistance than M14 cancer cells against hydrogen peroxide and aldehyde(s), generated by the enzymatic oxidation of spermine, able to cause cytotoxicity [1]. As shown in Figure 8, also, primary neurons are more resistant against BSAO/SPM treatment than both NB cell lines, SJNKP and IMR5. In performing the experiments, to compare NB cells with primary neurons, these primary cells were not detached prior to the treatment, as usually happens, but they were treated with BSAO/SPM in multiwell plates, in adherent condition. This different treatment might cause any changes represented in the cytotoxic effect between Figure 1 and Figure 8. It was necessary to perform the experiments in this way since primary neurons, if detached, are not able to attach again on multiwell plates as requested by both clonogenic and MTT assays. However, the treatment with BSAO/SPM induced a higher cytotoxic effect on both NB cell lines than in primary neurons isolated from rats. The mechanism of this interesting result, due to the different effect in cytotoxicity between NB cells and primary neurons after the treatment with BSAO/SPM, still requires further investigations to be explained. However, we hypothesize that the higher activity of the intracellular defense systems in both melanocytes and neurons is able to remove the cytotoxic effect induced by H_2_O_2_ and aldehyde(s), conferring a higher resistance to both cell lines, melanocytes and neuron cells, than to cancer cells, against BSAO/spermine-induced cytotoxicity [1]. Taken together, our data sustain the concept that this therapeutic approach deserves to be validated in vivo in order to verify whether BSAO/SPM decreases tumor growth in xenograft mouse models using the cell lines employed in the present study.

## 5. Conclusions

In conclusion, this study showed that the treatment with BSAO and spermine induced cytotoxicity through a mitochondrial-dependent apoptotic mechanism in NB cell lines. Analysis of mRNAs and proteins also showed that the treatment caused upregulation of proapoptotic mRNAs, miRs, and proteins. Furthermore, the cytotoxicity was greater in NB cell lines than in primary neurons. Although these experiments represent only the first step to a potential clinical application, and a further work is still required, the results described are very promising and suggest that, based on these observations, it could be possible to realize a new therapeutic antitumor approach.

## Figures and Tables

**Figure 1 cells-10-01950-f001:**
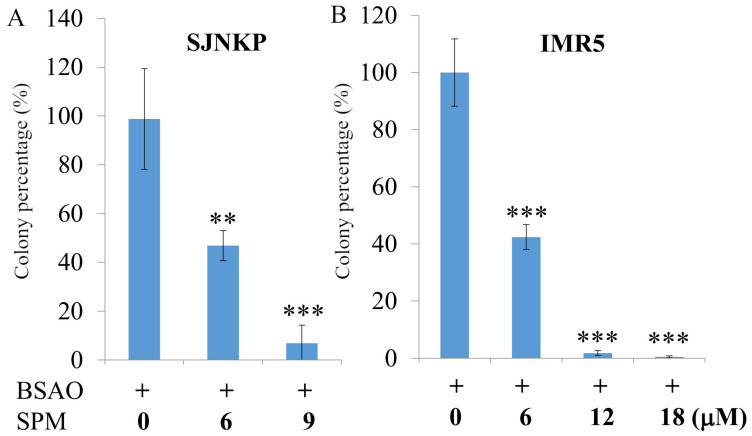
Cytotoxic effects induced by BSAO and SPM on SJNKP and IMR5 cell lines. After treatment with BSAO and SPM for 1 h, SJNKP and IMR5 cells were plated on Petri dishes or 96-well plates and incubated at 37 °C for 13 days or 48 h, respectively. In the clonogenic assay (**A**,**B**), the medium was discarded at the end of the treatment, the colonies were washed with PBS, fixed in 70% ethanol for 10 min, stained with methylene blue for 10 min, and counted. Each point represents the mean ± SEM of three independent experiments, with two to five plates per experiment. Conversely, in MTT assay (**C**,**D**), 10 μL of MTT (5 mg/mL) were added. After 3 h of incubation at 37 °C, the medium was discarded, and the formazan crystals formed in the cells were dissolved in 100 μL of DMSO for 30 min. Each point represents the mean ± SEM of four independent experiments, with three wells per experiment. Data are presented as mean ± SD. The statistical significance of differences between groups was determined by one-way analysis of variance followed by Dunnett’s post hoc multiple comparison tests. ** *p* < 0.01; *** *p* < 0.001. BSAO: bovine serum amine oxidase; SPM: spermine.

**Figure 2 cells-10-01950-f002:**
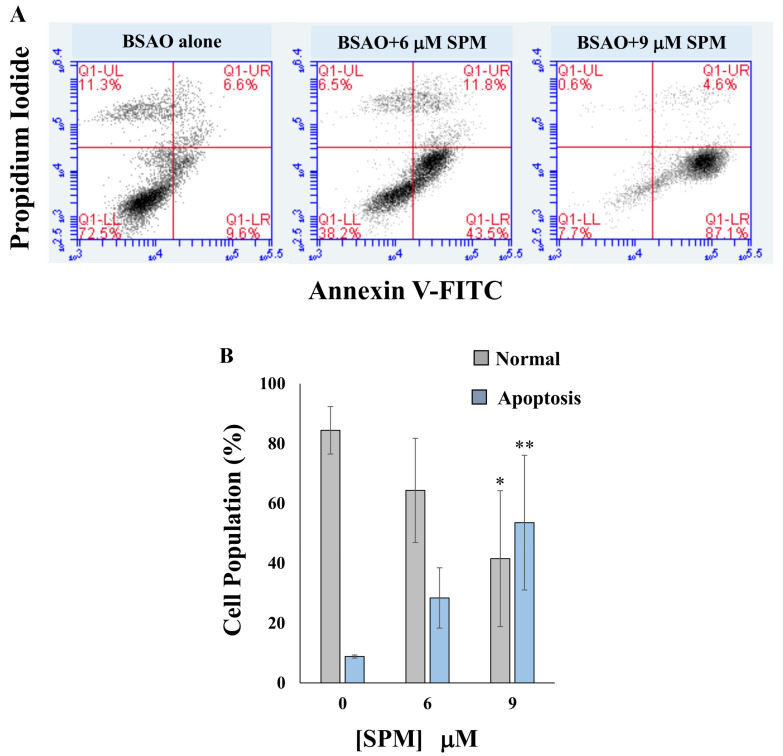
Increase of apoptosis after culturing SJNKP (**A**,**B**) and IMR5 (**C**,**D**) cell lines in the presence of BSAO and SPM. After the exposure to BSAO and SPM, SJNKP and IMR5 cells were seeded on multiwell plates and incubated for 48 h at 37 °C. Medium was collected and cells were detached by 10 mM EDTA and 0.1% trypsin, washed with PBS, and divided into two fractions. One was used for annexin V-FITC assay, the second one to detect cell cycle (in Figure 3). For annexin V-FITC and PI double staining assay, the cells were resuspended in binding buffer (10 mM HEPES/NaOH, pH 7.4, 140 mM sodium chloride, 2.5 mM CaCl_2_), and incubated with annexin V-FITC and PI for 10 min at room temperature. Annexin V-FITC and PI were analyzed on the FL1 and FL3 channels, respectively as reported by Kanamori et al. [50]. (**A**,**C**) Dot plots of SJNKP and IMR5 cells were analyzed using flow cytometry; the dot plot profiles of cells were obtained from one out of three independent experiments, performed in the same experimental conditions, which yielded similar results. (**B**,**D**) Quantitative analysis of percentage of normal and apoptotic cells in SJNKP and IMR5 cell lines. (**B**,**D**) Each bar represents the mean ± SD of normal or total apoptotic cells of three independent experiments. Where not shown, error bars lie within symbols. The statistical significance of differences between groups was determined by one-way analysis of variance followed by Dunnett’s post hoc multiple comparison tests. * *p* < 0.05; ** *p* < 0.01; *** *p* < 0.001. BSAO: bovine serum amine oxidase; SPM: spermine.

**Figure 3 cells-10-01950-f003:**
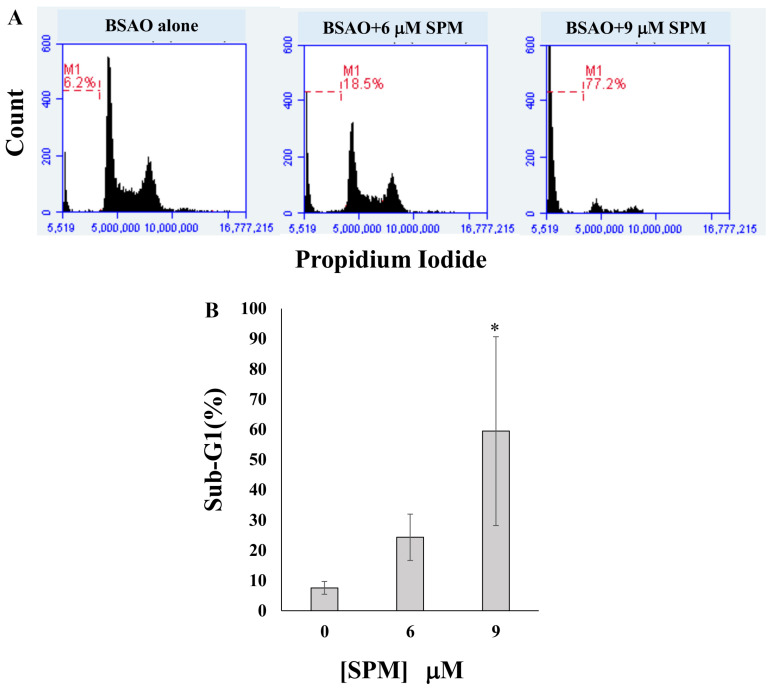
Cell cycle effects of culturing SJNKP (**A**,**B**) and IMR5 (**C**,**D**) cells in the presence of BSAO and SPM. After the exposure to BSAO and SPM, SJNKP and IMR5 cells were seeded on Petri dishes and incubated for 48 h at 37 °C. The medium was collected, and cells detached by 10 mM EDTA and 0.1% trypsin were washed with PBS (the second fraction as described in Figure 2). For cell cycle analysis, cells were fixed in cold 70% ethanol at 4 °C for 1 h. After washing with PBS, cells were incubated with PI and RNase A for 1 h at 37 °C in order to stain the nuclei. Cells were then subjected to flow cytometric analysis to examine the cell cycle. (**A**,**C**) Representative histograms of sub-G1 analysis performed on SJNKP and IMR5 cells using PI staining are shown. The percentage of the sub-G1 cell population is indicated. The histograms were obtained from one out of three experiments performed in the same experimental conditions, which gave similar results. (**B**,**D**) Quantitative analysis of sub-G1 phase of both SJNKP and IMR5 cells. Each bar represents the mean ± SD of sub-G1 cell population of three independent experiments. The statistical significance of differences between groups was determined by one-way analysis of variance followed by Dunnett’s post hoc multiple comparison tests. * *p* < 0.05; *** *p* < 0.001. BSAO: bovine serum amine oxidase; SPM: spermine.

**Figure 4 cells-10-01950-f004:**
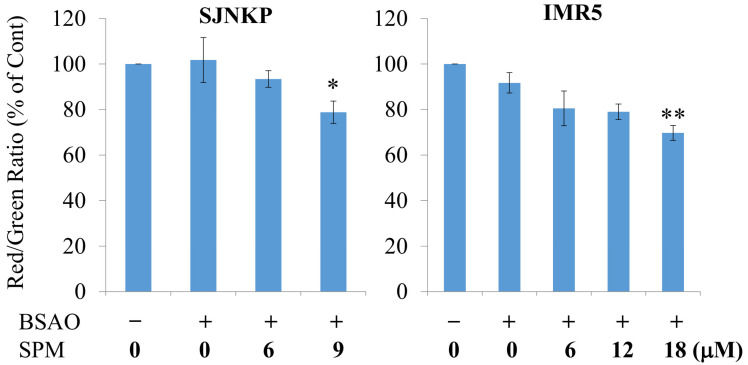
Mitochondrial depolarization was induced by BSAO and SPM in neuroblastoma cells. After incubation of the cells with BSAO and SPM for 1 h, followed by two washes, the red/green ratio was immediately detected by flow cytometry. JC-1 was added in the last 15 min of treatment, at a final concentration of 2.5 μg/mL. Data represent the mean values ±SD from three independent experiments. Where not shown, error bars lie within symbols. The statistical significance of differences between groups was determined by one-way analysis of variance followed by Dunnett’s post hoc multiple comparison tests. * *p* < 0.05; ** *p* < 0.01. BSAO: bovine serum amine oxidase; SPM: spermine.

**Figure 5 cells-10-01950-f005:**
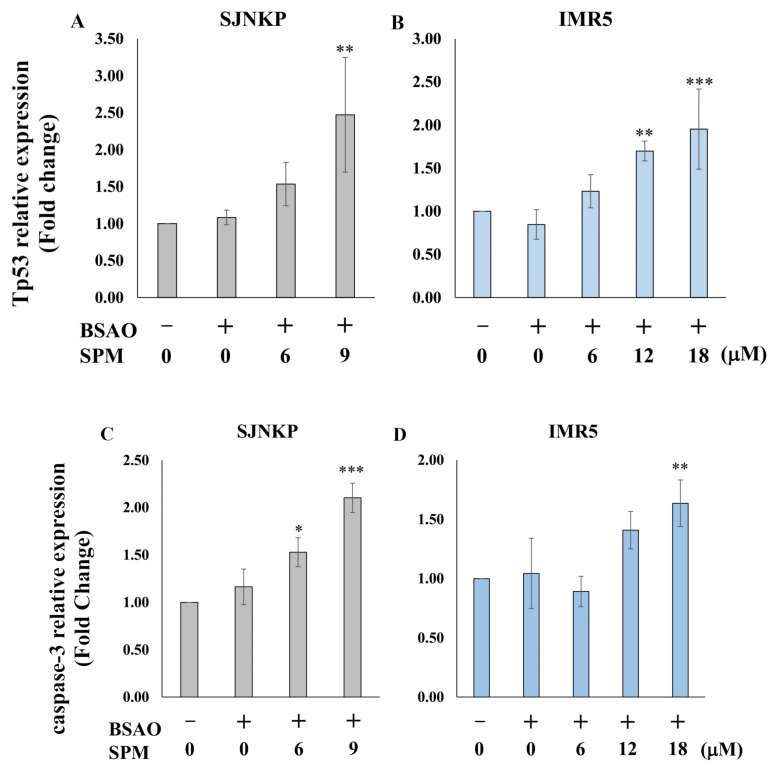
Analysis of mitochondrial apoptosis-related mRNAs. After incubation for 48 h with BSAO/SPM treatment, neuroblastoma cells were lysed by Tri reagent for RT-PCR analysis. Tp53 and caspase-3 mRNA levels were assayed by RT-qPCR in SJNKP (**A**,**C**) and IMR5 (**B**,**D**) cell lines. Data are presented as mean ±SD. RT-qPCR data were obtained in triplicate from four independent experiments. The statistical significance of differences between groups was determined by one-way analysis of variance followed by Dunnett’s post hoc multiple comparison tests. * *p* < 0.05; ** *p* < 0.01; *** *p* < 0.001. BSAO: bovine serum amine oxidase; SPM: spermine.

**Figure 6 cells-10-01950-f006:**
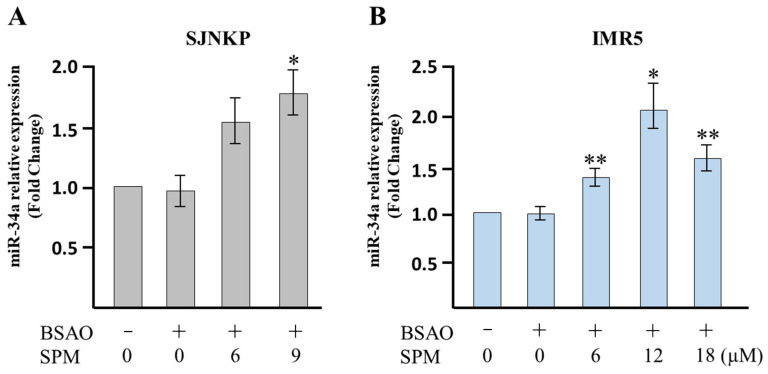
Analysis of apoptosis-related miR-34a. After the incubation for 48 h with BSAO/SPM treatment, neuroblastoma cells were lysed by Tri-Reagent for real-time RT-qPCR. (**A**) SJNKP cell line; (**B**) IMR5 cell line for all data; miR-let-7c and U6 snRNA were used as references. Data are presented as fold changes with respect to untreated cells and represent the mean ±SD. BSAO: bovine serum amine oxidase; SPM: spermine. The data represent seven (SJNKP) and five (IMR5) replicates from three independent experiments. The statistical significance of differences between groups was determined by one-way analysis of variance followed by Dunnett’s post hoc multiple comparison tests. * *p* < 0.05, ** *p* < 0.01 vs. BSAO only treated cells (control group).

**Figure 7 cells-10-01950-f007:**
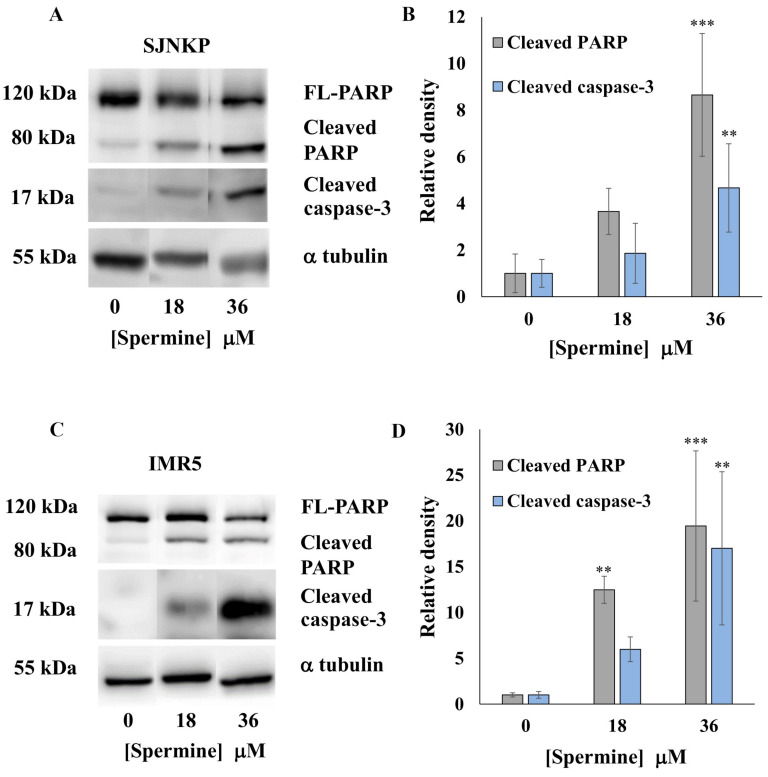
Expression of PARP and caspase-3 in neuroblastoma cell lines. Following incubation with BSAO/SPM treatment for 48 h, neuroblastoma cells were lysed in RIPA buffer containing proteases inhibitors for Western blotting analysis. (**A**,**C**) Representative Western blot of full-length PARP and cleaved PARP and caspase-3 protein from SJNKP (**A**) and IMR5 (**C**) cell lines. α-tubulin was used as a loading control. (**B**,**D**) Quantification (relative density) of the intensity of staining of full-length PARP and cleaved PARP and caspase-3 protein from SJNKP (**B**) and IMR5 (**D**) cell lines detected by Western blotting. Data are expressed as mean ±SD of four independent experiments, each performed in triplicate. The statistical significance of differences between groups was determined by one-way analysis of variance followed by Dunnett’s post hoc multiple comparison tests. ** *p* < 0.01, *** *p* < 0.001 vs. control group. FL indicates full length.

**Figure 8 cells-10-01950-f008:**
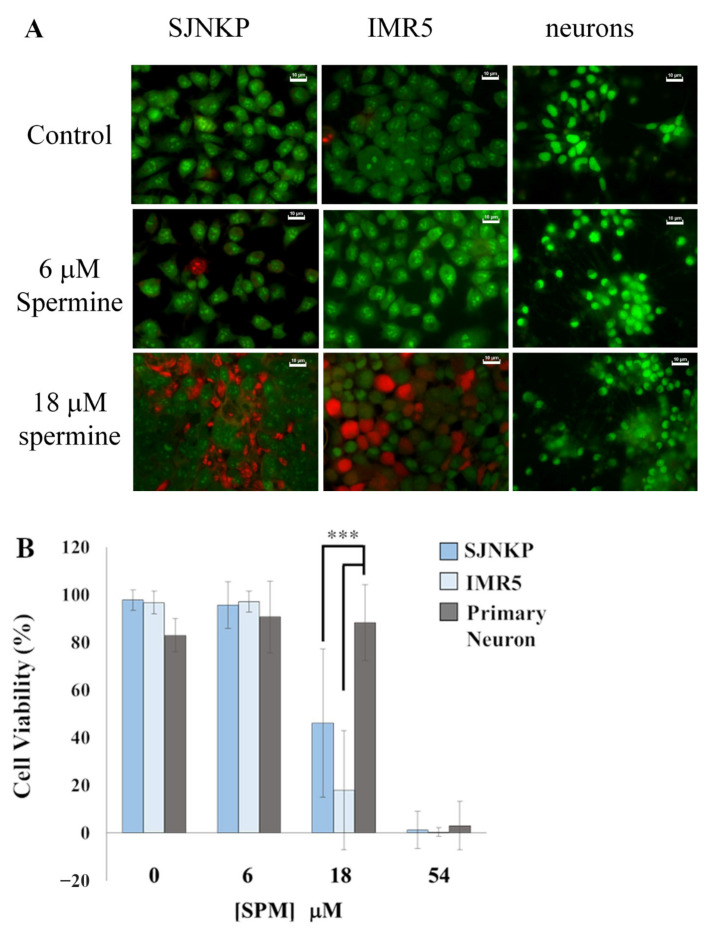
Apoptosis is evidenced by AO/EB double staining in neuroblastoma cells and neurons. Each cell type was cultured on at least four different round glasses for microscopy; 48 h after the treatment with BSAO/SPM, cells were stained with AO/EB to detect apoptosis. Fluorescence images were taken from all the glasses in random fields, and calculation of viable cell number was carried out for each cell type after dividing each image into quarters. Percentages of viable cell were calculated by ImageJ. (**A**) Fluorescent microscope pictures. Viable cells excluded ethidium bromide and appear bright green, while apoptotic cells are orange to red in color, bar = 10 µM. (**B**) Percentages of viable cells were calculated by ImageJ. Values are means ± SD. The statistical significance of differences between groups was determined by one-way analysis of variance followed by Dunnett’s post hoc multiple comparison tests. *** *p* < 0.001. SPM indicates spermine.

**Figure 9 cells-10-01950-f009:**
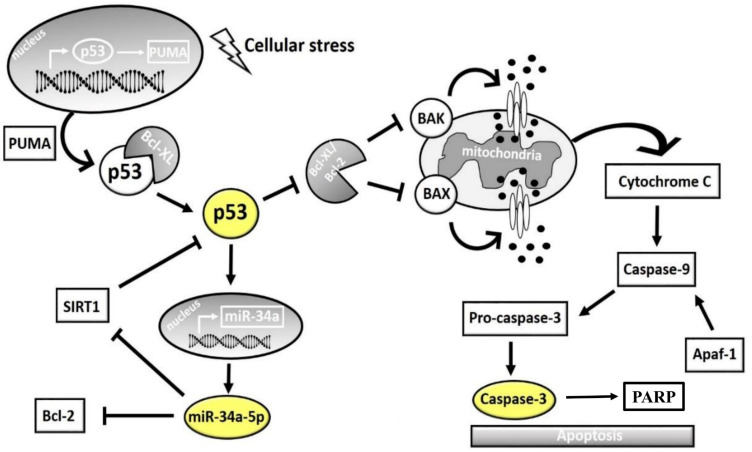
Interplay among Tp53, caspase-3, SIRT1, BCL2, and miR-34a. Cellular stress induces p53 expression and its translocation from nucleus to cytoplasm, mediated by PUMA. p53 interacts with antiapoptotic BCL2 and BCL-XL proteins, activating the proapoptotic proteins BAK and BAX located on mitochondrial membrane. BAX- and BAK-mediated creation of pores in the mitochondrial membrane activates cytochrome C release that results in activation of caspase-9, which in turn activates the effector caspase-3. Moreover, p53 is able to regulate the proapoptotic microRNA, miR-34a, which in turn targets the 3′UTR of SIRT1 mRNA, resulting in its downregulation. At the same time, SIRT1 represses p53 expression, creating a feedback loop.

**Table 1 cells-10-01950-t001:** List of employed TaqMan assays for analysis of transcripts. Assays for mRNA analysis were purchased from Integrated DNA Technology (IDT).

Assays for Transcripts Expression Analysis		
Gene Name		Probe Sequences
*Tp53*	Probe	5′-/56FAM/TCCCAGAAT/ZEN/GCAAGAAGCCCAGA/3IABkFQ/-3′
	Primer FW	5′-AACCCACAGCTGCACAG-3′
	Primer RW	5′-CCTTCCCAGAAAACCTACCAG-3′
*caspase-3* *RPL13A*	Probe	5′-/56-FAM/AGTTTCGTG/ZEN/AGTGCTCGCAGCTC/3IABkFQ/-3′
	Primer FW	5′-CACGGATACACAGCCACAG-3′
	Primer RW	5′-CGGATGGGTGCTATTGTGAG-3′
	Probe	5′-/5HEX/CGCACGGTC/ZEN/CGCCAGAAGAT/3IABkFQ/-3′
	Primer FW	5′-GGCAATTTCTACAGAAACAAGTTG-3′
	Primer RW	5′-GTTTTGTGGGGCAGCATACC-3′

**Table 2 cells-10-01950-t002:** List of employed TaqMan assays for analysis of miRNA. Assay for miRNA analysis were obtained by Applied Biosystem.

Assays for miRNAs Expression Analysis		
miRNA Name	Cat. Number	Assay ID
miR-34a-5p	4427975	000426
miR-let-7c-5p	4427975	000379
snRNA U6	4427975	001973

## Data Availability

Data availability statements were all added in the whole text, in the template. No supplementary materials are enclosed.

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
