# Peer review of "Enzymatic Spermine Metabolites Induce Apoptosis Associated with Increase of p53, caspase-3 and miR-34a in Both Neuroblastoma Cells, SJNKP and the N-Myc-Amplified Form IMR5"

_cells, 2021, doi:10.3390/cells10081950_

Round 1

Reviewer 1 Report

Minor: 

Figure 2 legend: line 469, please check “*p < 0.05; **p < 0.01; **p<0.001”

Figure 3  legend: line 485, please check  “*p < 0.05; **p <  0.01; **p<0.001.”. There is no “**” labeled on Figure 3.

Figure 5 legend: line 537, please check “*p < 0.05; **p < 0.01; **p<0.001.”.

Figure 6 legend: line 562, “*P < 0.05, **P < 0.01, ***P< 0.001”.  There is no “***” labeled on Figure 6.

Author Response

Object: Manuscript ID: cells-1310952 -  Comments’ REVIEWER 1

Manuscript “Enzymatic spermine metabolites induce apoptosis associated with increase of p53, Caspase-3 and miR-34a in both neuroblastoma cells, SJNKP and the N-Myc amplified form IMR-5”, by Kanamori et al.  

Answer to the Comments’ REVIEWER 1:

The Authors thank the reviewer for comments and precious suggestions to improve legends of Figures 2 – 3- 5  and 6. All corrections have been done as requested and highlighted in red in the text:

Figure 2 legend: line 469, we replaced “*p < 0.05; **p < 0.01; **p<0.001” with  *p < 0.05; **p < 0.01; ***p<0.001

Figure 3  legend: lines 485 and 486, we replaced  “*p < 0.05; **p <  0.01; **p<0.001.” with *p < 0.05; ***p<0.001 . Instead **p <  0.01 was removed since  there is no “**” labeled on Figure 3, as suggested by the reviewer.

Figure 5 legend: line 537, we replaced “*p < 0.05; **p < 0.01; **p<0.001.” with *p < 0.05; **p < 0.01; ***p<0.001

Figure 6 legend: line 562, we replaced  “*P < 0.05, **P < 0.01, ***P< 0.001” with *P < 0.05, **P < 0.01.  Instead ***P< 0.001 was removed since  there is no “***” labeled on Figure 6, as suggested by the reviewer.

Kind Regards

Reviewer 2 Report

The authors have answered to the first question, but all other questions still remain without convincing answers.

Question: Cell viability was decreased by about 50% in SJNKP and 70% in IMR5 cells when incubated with BSAO and 6 µM and 12 µM of SPM, respectively. However, the ratio of apoptotic cells versus normal cells at the same concentrations (Fig 2B and Fig 2D) is not supporting the viability data.

Authors answer: The following sentence has been included in the text, lines 426 and 427: “The effects of 6 µM SPM on BSAO treated cells (Fig.2, A and B) exhibit the same trend found in the cytotoxic assays described in Figure 1”.

Reviewer: The data presented in Fig2A and 2C support the viability data as indicated by the authors, however the data presented in Fig2B don't. There is no significant difference between cell population of normal cells in control compared with cells treated with 6 mM SPM, while apoptosis was about 43% when treated with 6 mM of SPM (Fig2A). There is still a problem with this figure.

Question: The author stated that BSAO/SPM induced cytotoxicity through a mitochondrial-dependent apoptotic mechanism, however mitochondrial depolarization occurs only at the highest concentration of SPM ie. 9 µM for SJNKP and 18 µM  for IMR5, whereas cell viability was already affected at 6 µM  in both cell lines.

Authors answer: We agree with the reviewer that this should be clearly stated. We suggest to add a brief sentence in the text, lines 508 and 509 “…. induce an evident Δψm dissipation, at least when the highest doses of SPM were employed, since …..”

Reviewer: I agree with this. But, if depolarization does not occur at 6 mM in SJNKP and 12 mM in IMR5 means that cell death observed (43% apoptosis in the case of SJNKP and 70% for IMR5) is not induced through a mitochondrial-dependent apoptotic mechanism as suggested by the authors. This suggestion stands true for higher SPM concentrations, if we assume that there is no other mechanism involved.

Question: The authors showed that about 50% of SJNKP and IMR5 cells (Fig 2B and Fig2D) presented apoptosis when treated with BSAO and SPM at 9 µM (SJNKP) and 12 µM (IMR5), respectively. However, cleavage of PARP and Caspase-3 was shown for higher concentrations of SPM ie.18 µM and 36 µM. PARP and Caspase-3 cleavage should be shown for the concentrations used in apoptosis analysis (Fig 2B and Fig 2D).

Authors answer: We further clarified the sentence in the text, at lines 703 and 704 “…. employed (compare for instance the experiments shown in Figure 7 with those described in Figures 2 and 3); still, it’s worth noting …..”.

This sentence did not respond to the question. Fig2A clearly showed that apoptosis occurs at high rate of SPM ie. 6 mM and 9 mM SPM in SJNKP (43% and 87%), whereas in Fig. 7 higher concentrations were used ie. 18 mM and 36 mM for studying the cleavage of PARP and Caspase 3. The effect of such higher concentrations was not evaluated neither in viability nor in apoptosis tests for these cells. In my point of view, it is important to show the cleavage of these apoptotic markers, if any, at the concentrations used for apoptosis study in Fig2A.

Author Response

Object: Manuscript ID: cells-1310952 - Answer to the Comments’ REVIEWER 2  

PLEASE SEE ALSO THE ATTACHMENT

Manuscript “Enzymatic spermine metabolites induce apoptosis associated with increase of p53, Caspase-3 and miR-34a in both neuroblastoma cells, SJNKP and the N-Myc amplified form IMR-5”, by Kanamori et al.  

The Authors thank the Reviewer for the comments to our paper.  

Answer to the Comments’ REVIEWER 2  

Reviewer: The data presented in Fig2A and 2C support the viability data as indicated by the authors, however the data presented in Fig2B don't. There is no significant difference between cell population of normal cells in control compared with cells treated with 6 mM SPM, while apoptosis was about 43% when treated with 6 mM of SPM (Fig2A). There is still a problem with this figure.

 Authors answer:  Sorry, we disagree. The data presented in Fig2B support Annexin V assay reported in Fig. 2A. There is significant difference between cell population of normal cells in control (about 8% apoptosis) compared with cells treated with 6 µM SPM (about 34%). This value is approximately similar to that obtained by assaying with Annexin V where apoptosis was about 43%, when cells were treated with 6 µM of SPM (Fig2A). These results are also supported by Figs, 3B and 3D cell-cycle that show the hypodiploid peak (Sub-G1 % peak). Therefore, the Authors don’t see any problem about this figure. It is not possible to obtain superimposable results between two different methods as the Reviewer wishes.

Reviewer: I agree with this. But, if depolarization does not occur at 6 mM in SJNKP and 12 mM in IMR5 means that cell death observed (43% apoptosis in the case of SJNKP and 70% for IMR5) is not induced through a mitochondrial-dependent apoptotic mechanism as suggested by the authors. This suggestion stands true for higher SPM concentrations, if we assume that there is no other mechanism involved.

Authors answer: Sorry but the reviewer's observation is not accurate. A slight depolarization occurs at 6 µM in SJNKP and at 6 and 12 µM  in IMR5 (Fig. 4). Instead was higher at 9  µM in SJNKP and at 18  µM in IMR5. In fact, in the previous revision the Authors added a brief sentence in the text, lines 508 and 509 : “ …. induce an evident Δψm dissipation, at least when the highest doses of SPM were employed, since …..”. About other mechanism involved, at present, the authors think that cell death is mainly  induced by an apoptotic mechanism as supported by Annexin V, cell-cycle, Western Blot…….assays. However, a very preliminary study, using DAPI and TUNEL methods and WB with antibody pAKT, has shown that a low spermine concentration cell death might be also induced by autophagy. However, this investigation is in progress and needs further experiments.  

Reviewer; This sentence did not respond to the question. Fig2A clearly showed that apoptosis occurs at high rate of SPM ie. 6 mM and 9 mM SPM in SJNKP (43% and 87%), whereas in Fig. 7 higher concentrations were used ie. 18 mM and 36 mM for studying the cleavage of PARP and Caspase 3. The effect of such higher concentrations was not evaluated neither in viability nor in apoptosis tests for these cells. In my point of view, it is important to show the cleavage of these apoptotic markers, if any, at the concentrations used for apoptosis study in Fig2A.

Authors answer: Sorry we disagree, since our sentence respond to the reviewer’s question. The Reviewer does not need to read only the sentence in the text, at lines 703 and 704, i.e. in the second review  “…. employed (compare for instance the experiments shown in Figure 7 with those described in Figures 2 and 3); still, it’s worth noting …..”, BUT the Reviewer must take into account also the sentence reported in the text in the first review, lines 701-703. Therefore,  he must read: “”We would like to underline that the BSAO/SPM effects occurring at different concentrations of reagents are largely expected in the different cellular assays employed (compare for instance the experiments shown in Figure 7 with those described in Figures 2 and 3); still, it’s worth noting a same trend was observed””, lines 701-705.

We wish to thank again the   Reviewer  for the comments. They were highly appreciated.

Kind Regards,

This manuscript is a resubmission of an earlier submission. The following is a list of the peer review reports and author responses from that submission.

Round 1

Reviewer 1 Report

  1. There are a lot of redundancies in the Results, some should belong to Introduction or methods.
  2. Figures: The number of samples should be indicated for Figures 1-8. In addition, with limited numbers of samples, **p < 0.01 and ***p<0.001 are not statistically meaningful.
  3. Legend of Figure 2: line 456 and 457, “divided into two 456 Figure 3”, please clarify.

Author Response

Manuscript ID: cells-1257281

Cover letter Reviewer 1 point by point: The Authors thank both Editor and Reviewers for their effort in reviewing the manuscript. The Authors agree with all their Comments and Suggestions, as reported in the text and comments  enclosed (rebuttal letter). They  have certainly improved the whole manuscript.

Reviewer 1

Comments and Suggestions for Authors:

Question.  There are a lot of redundancies in the Results, some should belong to Introduction or methods.

Answer. As suggested by the reviewer, several sentences, listed below, have been removed from the results and reported in other sections,  highlighted in red: 

- From results paragraph 3.1. Page 10,  lines 390 to 392, the following sentence was deleted because it was already present in Materials & Methods, paragraph  2.5  Clonogenic Assay:

Following the treatment with BSAO and SPM for 1 hour at 37°C, IMR5 and SJNKP cells were plated on Petri dishes and incubated at 37°C for 13 days until macroscopic colonies (≥ 50 cells) were formed.

- From results paragraph 3.1. Page 10, line, 397 to 399, the following sentence was deleted because it was already present  in Materials & Methods, paragraph  2.5 MTT  Assay:

Briefly, after treatment with BSAO and SPM for 1 hour at 37°C, IMR5 and SJNKP cells were plated on a 96-well plate and incubated for 48 hours, at 37°C.

- From results paragraph 3.2. Page 11,  lines 419 to 421, the following  sentence was deleted and then added to the session Materials & Methods, Page 6, paragraph  2.7, lines 258-262, highlighted in red: 

Annexin V-FITC/PI assay is a Ca2+‑dependent phospholipid‑binding protein which shows a high affinity for phosphatidylserine residues located on the outer layer of the plasma membrane cells. It represents an early signal of the apoptotic process.  

- From results paragraph  3.2. Page 11, lines 422-425,  the following  sentence was deleted because it was already present  in Materials & Methods, paragraphs  2.7 and 2.8 :

The results obtained by flow cytometric analysis are shown in Figures 2 and 3, respectively. After the exposure to BSAO and SPM, SJNKP and IMR5 cells were seeded on multiwell plates and incubated for 48 hours; then both cell lines were subjected to flow cytometric analysis.   

- From results paragraph  3.6. Page 19, lines 563-568,  the following sentence  was deleted and then added to the session  Materials & Methods, paragraph  2.12, Page 8, lines 343-348 highlighted in red:

Cleavage of Caspase 3 and of poly-ADP-ribose polymerase (PARP, a major substrate of caspase- 3) are considered valuable markers of apoptosis. Caspase-3 is a member of the caspase family that comprises 13 proteases that play a central role in the activation of the apoptotic program. The activation of caspase-3 mediates apoptotic cell death through the downstream cleavage of several key cytoplasmic or nuclear substrates and is primarily responsible for the cleavage of PARP that signals DNA damage leading to cell death.   

- From results paragraph  3.7. Page 20, lines 585-589-   the following sentence  was deleted and then added  in Discussion, Page 24,  lines 705-709, highlighted in red :

When studying the potential usefulness of a given treatment, it is of the most importance to compare the effects of the compounds under study on both cancer and normal cell. In this study were performed experiments to verify the selectivity of the BSAO/SPM treatment for tumor cells excluding toxic phenomena on normal cells, representative of healthy tissues.    

Question.  Figures: The number of samples should be indicated for Figures 1-8. In addition, with limited numbers of samples, **p < 0.01 and ***p<0.001 are not statistically meaningful.

Answer. Sorry, this information was casually omitted by the authors. Now, the number of samples has been indicated for all the Figures,  from 1 to 8,  and highlighted in red :   Figure 1, Page 11,  lines 410-411 and 413-414 ”… For clonogenic assay (A and B): Each point represents the mean ± SEM of three independent experiments, with 2 to 5 plates per experi­ment; For MTT assay (C and D): Each point represents the mean ± SEM of four independent experiments, with 3 wells per experiment…” Figure 2, Page 14, lines 462 to 466 ”….. the dot plots profile of cells were obtained from one out of three independent experiments, performed in the same experimental conditions, which yielded similar results”.   ”….B and D) Each bar represents the mean ± SD of normal or total apoptotic cells of three independent experiments. Where not shown, error bars lie within symbols….”  Figure 3, Pages 15 and 16, lines 478-482.”..….The percentage of the sub‑G1 cell population is indicated. The histograms have been obtained from one out of three experiments performed in the same experimental conditions, which gave similar results. B and D) Quantitative analysis of Sub-G1 phase of both SJNKP and IMR5 cells. Each bar represents the mean ± SD of sub‑G1 cell population of three independent experiments….” Figure 4, Page 16, lines 512-513. “…...Data represent the mean values ± SD from three independent experiments. Where not shown, error bars lie within symbols…..” Figure 5, Page 18, lines 533-534. “......RT-qPCR data were obtained in triplicate from four independent experiments….”  Figure 6, Page 19, lines 558-559. “……The data represent 7 (SJ-NKP) and 5 (IMR-5) replicates from three independent experiments……”  Figure 7, Page 20, line 581. “......Data are expressed as mean ± SD of four independent experiments, each performed in triplicate….”  Figure 8, Page 21, lines 602-60  “......Each cell type was  cultured  on at least four different round glasses for microscopy; 48 hours after the treatment with BSAO/SPM, cells were stained with AO/EB to detect apoptosis. Fluorescence images were taken from all the glasses in random fields, and calculation of viable cell number  was carried out, for each cell type, after dividing each image into quarters. Percentages of viable cell were calculated by  Image J. ….”

Question.  Legend of Figure 2: line 456 and 457, “divided into two 456 Figure 3”, please clarify.

Answer. Sorry, something happened during the submission of the manuscript. In our copy the Authors already wrote: “”...... cells were detached by 10 mM EDTA and 0.1% trypsin, washed with PBS and divided into two fractions. One was used for Annexin V-FITC assay, the second one to detect cell cycle (in Figure 3).”” However, now, in Fig. 2 this sentence has been added to the file Word received, Page 13, lines 457 and 458,  highlighted in red.

 The Authors thank for your kind cooperation. It is greatly appreciated.

Best regards,

Enzo Agostinelli

Reviewer 2 Report

The manuscript by Y. Kanamori et al., entitled “Enzymatic spermine metabolites induce apoptosis associated with increase of p53, caspase-3 and miR-34a in both neuroblastoma cells, SJNKP and the N-Myc amplified form IMR-5“ aimed to study the effect of incubation of neuroblastoma cells in culture with bovine serum amino oxidase (BSAO) and spermine (SPM). The authors showed that incubation with BSAO/SPM reduced cell viability, induced apoptosis and sub-G1 cell accumulation in both cell lines. They reported that BSAO/SPM treatment induced mitochondrial membrane depolarization, increased the expression of pro-apototic gene p53, Caspase-3 and miR34a. The authors also showed the cleavage of PARP and Caspase-3. Finally, they reported that neurones were more resistant to the cytotoxic effects of BSAO/SPM treatment than neuroblastoma cancer cells.

This study is similar to that performed in LoVo human colon adenocarcinoma cells by S. Ohkubo et al. in 2019 and published in Int J Oncol using BSAO and Maize polyamine oxidases and SPM.

This paper outlines a series of experiments that appear logical. However, there are some points that need to be completed and clarified.

Major points:

  • The authors showed that BSAO/SPM induced cytotoxicity in neuroblastoma cells in vitro, it would be interesting to examine whether BSAO/SPM decreases tumor growth in a xenograft mouse model using both cell lines.

  • The authors indicated that BSAO/SPM induced cytotoxicity through a mitochondrial-dependent apoptotic mechanism. To gain more insight in the mechanism involved, it is essential to identify the metabolites involved and their targets.

To be clarified:

The authors showed that BSAO alone did not produce cytotoxic effects on both SJNKP and IMR5 cells, however the cytotoxic effect of SPM alone, if any, at different concentrations used or at least at the highest concentration has not been examined.

Cell viability was decreased by about 50% in SJNKP and 70% in IMR5 cells when incubated with BSAO and 6 mM and 12 mM of SPM, respectively. However, the ration of apoptotic cells versus normal cells at the same concentrations (Fig 2B and Fig 2D) is not supporting the viability data.

The author stated that BSAO/SPM induced cytotoxicity through a mitochondrial-dependent apoptotic mechanism, however mitochondrial depolarization occurs only at the highest concentration of SPM ie. 9 mM for SJNKP and 18 mM for IMR5, whereas cell viability was already affected at 6 mM in both cell lines.

The authors showed that about 50% of SJNKP and IMR5 cells (Fig 2B and Fog2D) presented apoptosis when treated with BSAO and SPM at 9 mM (SJNKP) and 12 mM (IMR5), respectively. However, cleavage of PARP and Caspase-3 was shown for higher concentrations of SPM ie.18 mM and 36 mM. PARP and Caspase-3 cleavage should be provided for the concentrations used in apoptosis analysis (Fig 2B and Fig 2D).

Author Response

Manuscript ID: cells-1257281

 Cover letter Reviewer 2 point by point: The Authors thank both Editor and Reviewers for their effort in reviewing the manuscript. The Authors agree with all their Comments and Suggestions, as reported in the text and comments  enclosed (rebuttal letter). They  have certainly improved the whole manuscript. PLEASE SEE ALSO THE ATTACHMENT 

Reviewer 2

Comments and Suggestions for Authors:

The manuscript by Y. Kanamori et al., entitled “Enzymatic spermine metabolites induce apoptosis associated with increase of p53, caspase-3 and miR-34a in both neuroblastoma cells, SJNKP and the N-Myc amplified form IMR-5“ aimed to study the effect of incubation of neuroblastoma cells in culture with bovine serum amino oxidase (BSAO) and spermine (SPM). The authors showed that incubation with BSAO/SPM reduced cell viability, induced apoptosis and sub-G1 cell accumulation in both cell lines. They reported that BSAO/SPM treatment induced mitochondrial membrane depolarization, increased the expression of pro-apototic gene p53, Caspase-3 and miR34a. The authors also showed the cleavage of PARP and Caspase-3. Finally, they reported that neurones were more resistant to the cytotoxic effects of BSAO/SPM treatment than neuroblastoma cancer cells.

Reviewer. This study is similar to that performed in LoVo human colon adenocarcinoma cells by S. Ohkubo et al. in 2019 and published in Int J Oncol using BSAO and Maize polyamine oxidases and SPM.

Authors. The Authors thank the Reviewer for this comment that allows  to clarify the aim of this research. The first aim of the present  study was to determine whether this new approach can be proposed for other tumoral cell systems, such as neuroblastoma cancer cells, therefore extending and further validated previous works [see refs. 1,15].  

The study performed  by S. Ohkubo et al. in 2019 was carried out  using an enzyme purified from plant, a  polyamine oxidases, in the presence of spermine for inducing cytotoxicity  in LoVo human colon adenocarcinoma cells. While, in this study has been used an animal amino oxidase purified from bovine serum (BSAO). ZmPAO was used  due to both its higher catalytic efficiency and the lower molecular weight (53 kDa) compared to BSAO (170 kDa), which are important aspects for the delivery of these enzymes into tumor cells, for deaminating endogenous polyamines inducing so a killing selective on cancer cells, as a new anticancer future therapy. Therefore, as reported in Introduction highlighted in red, for these aims, in the present study using both NB cell lines, we furthermore  considered novel approaches focusing on the study of changes of gene expression, including the analysis of the expression of a microRNA associated to the apoptotic pathway. Following this comment a sentence was clarified in Introduction:

Introduction Page 3 highlighted in red,  lines from 133 to 137

Reviewer. This paper outlines a series of experiments that appear logical. However, there are some points that need to be completed and clarified.

Major points:

- Question.  The authors showed that BSAO/SPM induced cytotoxicity in neuroblastoma cells in vitro, it would be interesting to examine whether BSAO/SPM decreases tumor growth in a xenograft mouse model using both cell lines.

- Answer. As reported in Discussion (page 24, lines 712 - 713), anti-tumoral activities of native and immobilized BSAO on PEG have been previously demonstrated performing experiments in vivo using a mouse melanoma model [see refs. 70,71].  Antitumor treatments consisted of a single injection of enzyme into the tumor mass. When immobilized BSAO  was injected into the tumor mass, there was a marked decrease of 70 % of the tumor growth. This was compared with a decrease of  32 % of tumor size when the same amount of native BSAO was administered.  Therefore, all our data sustain the concept that this therapeutic approach deserves to be validated in vivo in order to verify whether BSAO/SPM decreases tumor growth in xenograft mouse models using both NB  cell lines employed in the present study. These experiments belonged to our previous protocol. Unfortunately, Covid-19 pandemic caused delay in performing this study. Therefore, following the reviewer's comment a sentence was added at the end of the Discussion:

Discussion Page 25, highlighted in red, starting  from line 734 to 736.

- Question.  The authors indicated that BSAO/SPM induced cytotoxicity through a mitochondrial-dependent apoptotic mechanism. To gain more insight in the mechanism involved, it is essential to identify the metabolites involved and their targets.

- Answer. As reported in Results Page 11, paragraph 3.2. lines 431-434, during the treatment with BSAO and SPM,  the percentage increase of cells in apoptosis is in function of spermine concentration. As previously determined by a fluorometric assay, the cytotoxicity is mainly due to the high concentration of H2O2 produced by the enzymatic reaction, during the first 10 minutes of the reaction (approximately 80%). Aldehyde contributed to the cytotoxic effect only for 20%. Moreover,  it was described by Calcabrini et al. 2002, H2O2 is able to cross both cellular membranes, and the inner membrane of mitochondria and directly interacts with endogenous molecules and structures, inducing an intense oxidative stress and damage under appropriate conditions, as observed by transmission electron microscopy (TEM) in LoVo colon adenocarcinomas cells [see ref. 49]. Therefore, the results obtained in this research suggest that spermine metabolites caused apoptosis, is anticipated by Δψm collapse in both SJNKP and IMR5 NB cell lines (Figure 4).

So, following the reviewer's comment the Authors just highlighted in red the last sentence already present in Discussion Page 23, lines 673-674.

 To be clarified:

- Question.  The authors showed that BSAO alone did not produce cytotoxic effects on both SJNKP and IMR5 cells, however the cytotoxic effect of SPM alone, if any, at different concentrations used or at least at the highest concentration has not been examined.

- Answer. The Authors reported, highlighted in red, in Results, paragraph 3.1. Page 10, lines 390-392  ““….. the treatment with BSAO alone was unable to cause cytotoxic effects. Also the treatment with SPM alone, in our experimental conditions (it also includes the highest concentration used, of 36 micromolar), didn’t induce cytotoxicity (data not shown)”". Since, as previously described by Agostinelli et al. (Life Chem Rep 1991;9:193–204), only spermine is able to induce at 340 micromolar less than 30% of cytotoxicity, the Authors  decided to lighten the graph inserting only the above sentence in the text. However, the authors are available to further reviewer's request, if necessary.  

- Question.  Cell viability was decreased by about 50% in SJNKP and 70% in IMR5 cells when incubated with BSAO and 6 mM and 12 mM of SPM, respectively. However, the ration of apoptotic cells versus normal cells at the same concentrations (Fig 2B and Fig 2D) is not supporting the viability data.

- Answer. The Authors explain this difference   considering   that   MTT   doesn't just reflect cells undergoing apoptosis but also those in proliferative arrest,  i.e. inhibition of cell proliferation, as the method is an index of metabolic activity.

- Question.  The author stated that BSAO/SPM induced cytotoxicity through a mitochondrial-dependent apoptotic mechanism, however mitochondrial depolarization occurs only at the highest concentration of SPM ie. 9 mM for SJNKP and 18 mM for IMR5, whereas cell viability was already affected at 6 mM in both cell lines.

- Answer. We thank the reviewer for this Question and also for the next one. Both questions allow the authors to clarify the concentration of SPM used in the different assays: We would like to underline that the BSAO/SPM effects occurring at different concentrations of reagents are largely expected in the different cellular assays employed and supported by the numerous experiments performed; in any case a same trend was observed.

This sentence, highlighted in red, was added in Discussion Page 23, lines 699-701.

- Question.  The authors showed that about 50% of SJNKP and IMR5 cells (Fig 2B and Fog2D) presented apoptosis when treated with BSAO and SPM at 9 mM (SJNKP) and 12 mM (IMR5), respectively. However, cleavage of PARP and Caspase-3 was shown for higher concentrations of SPM ie.18 mM and 36 mM. PARP and Caspase-3 cleavage should be provided for the concentrations used in apoptosis analysis (Fig 2B and Fig 2D).

- Answer. As reported above:

We would like to underline that the BSAO/SPM effects occurring at different concentrations of reagents are largely expected in the different cellular assays employed and supported by the numerous experiments performed; in any case a same trend was observed.

This sentence, highlighted in red, was added in Discussion Page 23, lines 699-701.

The Authors thank for your kind cooperation. It is greatly appreciated.

Best regards,

Enzo Agostinelli

Round 2

Reviewer 2 Report

The authors showed that BSAO alone did not produce cytotoxic effects on both SJNKP and IMR5 cells, however the cytotoxic effect of SPM alone, if any, at different concentrations used or at least at the highest concentration has not been showed.

Cell viability was decreased by about 50% in SJNKP and 70% in IMR5 cells when incubated with BSAO and 6 mM and 12 mM of SPM, respectively. However, the ratio of apoptotic cells versus normal cells at the same concentrations (Fig 2B and Fig 2D) is not supporting the viability data.

The author stated that BSAO/SPM induced cytotoxicity through a mitochondrial-dependent apoptotic mechanism, however mitochondrial depolarization occurs only at the highest concentration of SPM ie. 9 mM for SJNKP and 18 mM for IMR5, whereas cell viability was already affected at 6 mM in both cell lines.

The authors showed that about 50% of SJNKP and IMR5 cells (Fig 2B and Fog2D) presented apoptosis when treated with BSAO and SPM at 9 mM (SJNKP) and 12 mM (IMR5), respectively. However, cleavage of PARP and Caspase-3 was shown for higher concentrations of SPM ie.18 mM and 36 mM. PARP and Caspase-3 cleavage should be shown for the concentrations used in apoptosis analysis (Fig 2B and Fig 2D).